# Evaluation of Time-of-Flight Camera Positioning for AI-based Patient Pose Assessment in Radiography

## Abstract

In recent years, medical technology companies have increasingly been integrating Time-of-Flight cameras into their X-ray devices to support optimal collimation and patient positioning process. However, for many hospitals, it is not financially viable to acquire a new X-ray device with such a camera for those features. In order to still have support for e.g. patient positioning, without having to buy a new X-ray device and be dependent on proprietary algorithms, it is possible to only acquire a Time-of-Flight camera, attach it to the X-ray device, and use custom algorithms. In this work, we evaluated the ideal camera position for AI-supported patient pose assessment based on depth images for such a setup and assessed the usefulness of a setup with multiple cameras. For this, we generated a total of 461,550 synthetic depth images from CT data with 50 different camera positions and synthetic radiographs in order to investigate in 1,500 experiments how accurately patients' poses can be assessed from different camera positions. We found that a camera position perpendicular to the target anatomy being X-rayed is particularly well suited, and that adding a second camera does not improve performance.

## 1 Introduction

Incorrect positioning of the patient during the X-ray examination is one of the main reasons for the need of a retake due to the inadequate diagnostic quality of the resulting radiograph [1]. Retaking a radiograph implies avoidable additional exposure of the patient to radiation and costs the hospital time and money. Since furthermore the positioning process is not standardized and depends heavily on the patient and the experience of the radiographer, a system that automatically assesses the patient's pose before the radiograph is taken could help reduce or prevent retakes and ensure quality standards. In order to be able to potentially support the positioning process, medical technology companies have been integrating Time-of-Flight (ToF) cameras into their X-ray devices in recent years. However, only the fewest hospitals can afford to acquire a new X-ray device or carry out expensive upgrades for these features, and potentially pay for proprietary algorithms. A more cost-effective alternative, that still enables positioning support, is to purchase a ToF camera and integrate it into the positioning process along with custom algorithms for pose assessment. Furthermore, recent research shows that, for example, it is possible to attach ToF cameras to the X-ray device and assess patient poses based on depth images [2, 3]. For such a system, it is necessary to identify which position of the camera relative to the target anatomy to be X-rayed is ideal for assessing the pose. Since installing cameras in various positions in clinical practice would have been costly, we used a recently published framework in this work that can generate synthetic depth images and corresponding radiographs from CT scans [4]. Using this framework, we were able to generate a dataset of 461,550 synthetic depth images from 50 different camera positions and two zoom levels of upper ankle joints, and evaluated the camera positions in 1,500 experiments to assess their suitability for pose estimation.

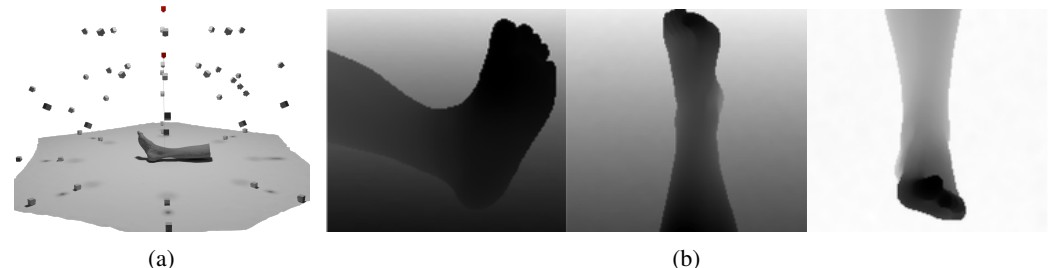

(a)                                                                    (b)

Figure 1: In 1a the different virtual camera positions are shown. The cameras marked in red indicate the camera positions at $(z = 0°, x = 0°)$ and thus perpendicular to the target anatomy. In 1b, three examples of synthetic depth images are shown from the following camera positions, left to right: $(z = 315°, x = 60°, zoom = 0)$, $(z = 180°, x = 30°, zoom = 0)$,$(z = 0°, x = 0°, zoom = 0)$.

We have also investigated the extent to which the use of multiple cameras can improve performance. Our experiments were conducted exclusively on upper ankle joints as the target anatomy, as they are difficult to position and, due to their anisotropic shape, are likely to have a greater impact on different camera positions than, for example, a knee. To the best of our knowledge, this is the first empirical evaluation of different camera positions for pose assessment.

## 2   Dataset and Methods

In order to evaluate a patient's pose based on depth images, radiographs taken together with corresponding depth images are mandatory, as only these radiographs allow for an assessment of diagnostic quality. Therefore, we applied the framework from Laufer et al.[4], as it enables generating synthetic depth images and the corresponding radiographs from CT scans, and expanded it to generate depth images of upper ankle joints from 50 different camera positions in two zoom levels. We chose the camera positions as spherical coordinates centered around the upper ankle joint, with a rotation around the global vertical z-axis in 45° increments (azimuth), where for a left foot 0° = distal, 90° = lateral, 180° = proximal, and 270° = medial, and a rotation around the x-axis (elevation) in 30° increments; see Figure 1a and Figure 2 for visualization. We have also distinguished between two zoom levels, whereby zoom level 1 is 30 cm closer to the target anatomy than zoom level 0. The generation process was analogous to Laufer et al.[4], resulting in a total of 461,550 synthetic depth images from CT scans of 10 patients with 17 upper ankle joints. The corresponding synthetic radiographs were labeled by four radiologists in Laufer et al.[4] regarding their diagnostic quality with a rating of 1 being the best and 3 being the worst. Radiographs with a rating above 2.5 can furthermore be classified as *non-diagnostic* and would have to be repeated in clinical practice. Note that the generation process takes into account the camera-specific intrinsic and extrinsic parameters, so that this dataset can also be generated for a specific ToF camera model. Figure 1b shows examples of the generated images.

In order to determine the ideal camera position for pose assessment, we conducted experiments for each of the 50 camera positions. The experiment design followed Laufer et al.[4] for comparability reasons and involved training an EfficientNet-B0 [5] with the depth images from one camera position as a regression with regard to the diagnostic quality of the corresponding radiographs to predict the quality of the pose. In order to be robust against outliers, three different testsets were defined for each camera position and each oft them trained with ten different seeds, resulting in 30 experiments per camera position, which amounts to a total of 1,500 experiments for all camera positions. The results of the different testsets and seeds were averaged for each individual camera position.

In order to further evaluate the extent to which the addition of another camera can improve pose assessment performance, we also evaluated combinations of two camera positions. For this, we averaged the predictions of identical images from the testset of two models trained with depth images from two different camera positions and evaluated them regarding the corresponding quality label. All experiments were evaluated with regard to the diagnostic quality, which indicates how many images were correctly predicted as *diagnostic* or *non-diagnostic*:

$$Diagnostic\ Accuracy = \frac{1}{N} \sum_{j=1}^{N} \mathbb{1}\left((y_j < 2.5 \wedge x_j < 2.5) \vee (y_j \geq 2.5 \wedge x_j \geq 2.5)\right),$$

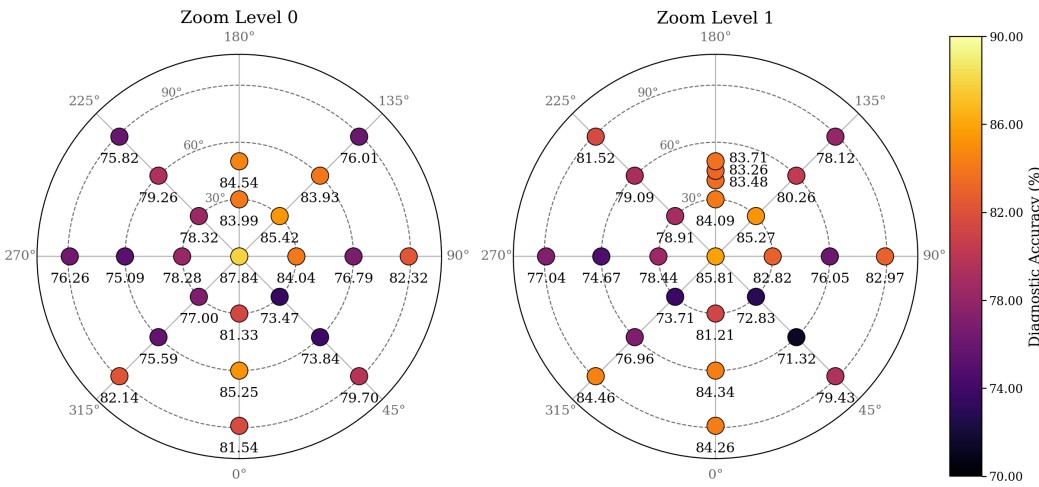

Figure 2: The Diagnostic Accuracy for pose assessment of the different camera positions and two zoom levels. The best camera position is the one perpendicular to the target anatomy.

where $y_j$ is the (average) prediction, $N$ is the number of samples, $x_j$ is the label, and $\mathbb{1}$ is the indicator function.

## 3 Results and Discussion

Figure 2 shows the results of the experiments and provides the average Diagnostic Accuracy for each of the 50 camera positions and both zoom levels. The results show that there are indeed camera positions such as $(z = 45°, x = 60°, zoom = 1)$ that should be avoided for pose evaluation due to their significantly poorer Diagnostic Accuracy. At both zoom levels, the best of all 50 camera positions is at $(z = 0°, x = 0°)$, i.e., directly vertical above the target anatomy. At zoom level 0, this position achieves a Diagnostic Accuracy of 87.84%. The large discrepancy of up to 17 percentage points between the different camera positions is also due to the anisotropic shape of the foot, which can lead to the occlusion of features that are potentially important for pose assessment from certain positions. A target anatomy with a more isotropic shape, such as that of the knee joint, could be expected to produce a more uniform distribution of accuracy across the various positions, which should be verified in future experiments. When comparing the different azimuth angles, the results show that the camera positions on the longitudinal axis at $z = 0°$ and $z = 180°$ perform best on average. Furthermore, it is visible that there is no significant difference between the two zoom levels for almost all camera positions. This is to be expected, as the different zoom levels essentially change the resolution of the images, which does not vary greatly over a distance of $30\,\mathrm{cm}$.

When combining two camera positions, the best result is achieved at position $(z = 0°, x = 0°, zoom = 0)$ and $(z = 135°, x = 60°, zoom = 0)$ with a Diagnostic Accuracy of 87.61%. It is possible that fusion strategies that use images from both camera positions already during training will result in a greater performance improvement between using one and two cameras, however, further experiments are required to verify this.

In this work, we were able to show, with the help of a synthetically generated dataset of depth images from 50 different camera positions, that pose assessment with a Diagnostic Accuracy of up to 87.84% is possible for the upper ankle joint. Out of these 50 different camera positions, we identified the camera position directly perpendicular above the target anatomy as the ideal position for pose assessment. This would make it suitable to attach a ToF camera directly to the X-ray device. Furthermore, due to the lack of performance improvement regarding the Diagnostic Accurcay when combining two cameras, an additional camera is not needed. With this work we have shown that hospitals and medical facilities can potentially benefit from features such as patient positioning assistance with a custom solution.

## Societal Impact Statement

The evaluation of ideal ToF camera positions for patient positioning assistance in radiography presented in this paper is intended to be a step toward independence from medical technology companies. Low resource hospitals and medical facilities should be able to create their own solutions for patient positioning assistance by purchasing a ToF camera and using their own data and algorithms, thereby improving patient care and avoiding the need for expensive upgrades or the purchase of new devices. This also ensures that older devices that are still functional can be used for longer, as new features can be upgraded inexpensively with custom solutions. However, further studies should be conducted to confirm the results presented in this paper, using a larger and potentially more diverse dataset and additional anatomies.

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
