# OpenReview forum: "Evaluation of Time-of-Flight Camera Positioning for AI-based Patient Pose Assessment in Radiography"
_EurIPS.cc/2025/Workshop/MedEurIPS — EurIPS 2025 Workshop MedEurIPS Submission_

### Official Review · Reviewer_Z6Gx · 2025-10-24
**Clear contribution with practical impact**

**Rating:** 8
**Confidence:** 4

**Review:**

The paper addresses a practical and well-motivated problem: rather than purchasing an expensive scanner with an integrated Time-of-Flight (ToF) camera, the authors investigate whether a separate ToF camera can be attached to an existing scanner and to help assess a patient’s pose using custom algorithms. The idea is interesting and should spark valuable discussions among the workshop's audience.
While the practical motivation is strong, the technical novelty appears incremental compared to prior work (e.g., citation [3] in the paper). I also encourage the authors to discuss the implications of using synthetic data (radiographs and depth images) for their experiments.
Overall, this is a solid workshop contribution that addresses a real clinical need, though clarifying the novel aspects would strengthen the paper.

---

### Official Review · Reviewer_GrZH · 2025-10-31
**Well motivated paper with lacking explanations**

**Rating:** 6
**Confidence:** 3

**Review:**

This paper presents a short study on selecting the best camera position for pose assessment in radiography. It is well motivated, putting emphasis on LMIC countries which could benefit from such an open approach. It presents a simple yet novel approach, which could potentially have impact on real clinical scenarios and foster discussion on how medical imaging could become more accessible. It contains two great figures that aid the reader in understanding the problem. Furthermore, they show a short analysis of how a second camera could influence the results.

However, it is lacking explanations and clarity - the authors heavily cite their previous work which hinders understanding of this paper

What could be improved:
- explain how the used pose estimation works
- explain the interplay between the 461,550 depth images and radiographs and camera angles
    - did each camera angle get multiple poses and radiographs?
    - was there a corresponding radiograph to each depth image?
    - does the orientation of the ankle on the table change?
    - it is unclear why there should be any differences between camera angles in terms of diagnostic accuracy - it seems that if the pose of the ankle changes then the best camera angle should also change

---

### Decision · Program_Chairs · 2025-11-03

**Decision:**

Accept (Poster)

**Comment:**

Both reviewers agree that the paper tackles a practical and relevant clinical problem and provides valuable insights for discussion at the MedEurIPS venue.